# A Predictive Guidance Obstacle Avoidance Algorithm for AUV in Unknown Environments

**DOI:** 10.3390/s19132862

**Published:** 2019-06-27

**Authors:** Juan Li, Jianxin Zhang, Honghan Zhang, Zheping Yan

**Affiliations:** 1Science and Technology on Underwater Vehicle Technology, Harbin Engineering University, Harbin 150001, China; 2College of Automation, Harbin Engineering University, Harbin 150001, China

**Keywords:** autonomous underwater vehicle, forward-looking sonar, predictive control, line-of-sight guidance, obstacle avoidance algorithm

## Abstract

A predictive guidance obstacle avoidance algorithm (PGOA) in unknown environments is proposed for autonomous underwater vehicle (AUV) that must adapt to multiple complex obstacle environments. Using the environmental information collected by the Forward-looking Sonar (FLS), the obstacle boundary is simplified by the convex algorithm and Bessel interpolation. Combining the predictive control secondary optimization function and the obstacle avoidance weight function, the predicting obstacle avoidance trajectory parameters are obtained. According to different types of obstacle environments, the corresponding obstacle avoidance rules are formulated. Lastly, combining with the obstacle avoidance parameters and rules, the AUV’s predicting obstacle avoidance trajectory point is obtained. Then AUV can successfully achieve obstacle avoidance using the guidance algorithm. The simulation results show that the PGOA algorithm can better predict the trajectory point of the obstacle avoidance path of AUV, and the secondary optimization function can successfully achieve collision avoidance for different complex obstacle environments. Lastly, comparing the execution efficiency and cost of different algorithms, which deal with various complex obstacle environments, simulation experiment results indicate the high efficiency and great adaptability of the proposed algorithm.

## 1. Introduction

Autonomous underwater vehicle (AUV) [1] is an important tool for marine resource exploitation and marine scientific research [2,3,4]. As more research interests turn to the cooperative target search, many factors should be considered, such as the environment information, target states, etc. In reality, the working environment for AUV is often unknown. In different types of obstacles, static or dynamic environments are encountered. Therefore, an effective obstacle avoidance algorithm is needed. 

In recent years, significant contributions have been made by many researchers in developing obstacle avoidance methods and applying them to various obstacle avoidance environments for AUV. These problems can be classified into the global obstacle avoidance path planning problems and local obstacle avoidance methods. When the global environmental information including various obstacles are known, the global obstacle avoidance path planning problem becomes a nonlinear optimal programming problem to find global optimal solutions on the pre-requisite that global variables are known. For example, two missile guidance algorithms are proposed for the intercept and for the rendezvous of a maneuvering target while avoiding a static obstacle by a specified avoidance distance [5]. However, the method simply solves the secondary optimization solution for static obstacles without considering the obstacle type and the impact of dynamic obstacles on the obstacle avoidance process [6,7]. According to the grid-based artificial potential field method, to solve the multi-mobile vehicle cooperative obstacle avoidance problem, it requires an effective method for managing general convex obstacles. However, it is easy to fall into local minimum value points when dealing with concave obstacles and failing to escape from concave obstacles. At the same time, the obstacle avoidance problem of the path planning is different from the calculation of obstacle avoidance trajectory points based on the real-time detection of obstacles. Control optimization problems for obstacle avoidance include non-linear programming [8,9], heuristic algorithm [10,11], and the graph search method including the A* algorithm [12,13,14] and D* algorithm [15]. In addition, if an AUV works in a locally known but globally unknown environment with various types of obstacles, nonlinear methods are needed to plan out the AUV trajectory points to ensure the safety of AUV in missions. To solve this problem, there is an artificial potential field method (APF) [16,17,18], as well as evolutionary algorithms such as the genetic algorithm (GA) [19,20,21] and the particle swarm optimization algorithm (PSO) [22,23]. Compared with traditional optimization methods, these algorithms usually lead to global optimal solutions, or approaches that are close to global optimal solutions. However, these evolutionary algorithms may cause poor numerical precision and difficult execution, when they solve the nonlinear optimal problem. At the same time, the iterative period of these algorithms is long. If not optimized, they will easily fall into the local minimum value.

Some significant achievements have been obtained in the obstacle avoidance problem. Masoud, Dadgar, et al. proposed an A-RPSO (Adaptive Robot PSO) algorithm [24], which considered the obstacle avoidance problem of the robot performing tasks, and, in the obstacle avoidance, there was also a control mechanism that escapes from the local optimum. In Reference [25], research studies are conducted on analyzing different trajectories presented by dynamic obstacles in the environment to predict their future positions and to realize obstacle avoidance. By estimating future areas where collisions between robots and obstacles may occur, mobile robots can take corrective actions before collisions. The Montegrey, Calif. AUV Research Naval Postgraduate Institute conducted an experimental study on underwater reactive obstacle avoidance (OA) for AUVs, by mainly focusing on using the ARIES AUV and Blueview Blazed Array FLS for obstacle detection and avoidance [26]. In References [27,28], an improved line-of-sight (LOS) guidance algorithm is used for obstacle avoidance. At the same time, in the process of underwater obstacle avoidance, to achieve the optimal path obstacle avoidance [29,30], the path planning method was used. For example, to achieve path smoothness, Joono, Sur. et al. [31] used a streamline-based autonomous underwater vehicle obstacle avoidance path planning method. Yufei, Zhuang, et al. [32] combined the particle swarm optimization (PSO) algorithm with the Legendre pseudo-spectral method (LPM), which achieved real-time collision avoidance of static obstacles and moving obstacles with different levels of positional uncertainty. Zheping, Y. et al. only classified the obstacles in the obstacle-avoiding environment, and did not consider the influence of the unevenness of the obstacles on obstacle avoidance. At the same time, the real-time obstacle avoidance method mentioned for the obstacle with uneven surface needs to calculate the turning radius when a large number of obstacle avoidance positions are calculated. In the case of multi-obstacle distribution, the obstacle data of the overlapping portion is not processed, but it is unreasonable to directly propose an obstacle avoidance gap for the overlapping obstacles as the direction of the obstacle avoidance. There is no comprehensive consideration of multiple factors affecting AUV obstacle avoidance. However, the previously mentioned methods are developed either for specific cases where different numbers of obstacles are distributed in the environment, or aimed at the obstacle and the AUV relative model to obtain the optimal solution in order to achieve AUV obstacle avoidance. No solution regarding complex irregular obstacles or obstacles of different types is proposed, and no optimization obstacle avoidance strategy is suggested for completely unknown environments. 

Based on the previously mentioned obstacle avoidance problems, and combining target search and tracking in unknown underwater environments with complex irregular obstacles, an obstacle avoidance method for AUV based on PGOA is proposed. The main idea of this algorithm is: The FLS (Forward-looking sonar) is equipped in the AUV front to obtain the obstacle information. Obstacles are classified according to the detected character information. Then, the contour convex algorithm and Bezier interpolation are used to change the irregular contour into a convex polygon, which simplifies the boundaries of the irregular complex obstacle, so that the AUV can respond quickly to various obstacle environments. In the second stage, by using the maximum turning radius calculation method based on the obstacle type and boundary data, different turning radii are obtained. Then, by using obstacle avoidance parameters and obstacle avoidance rules, the obstacle avoidance weight function is established. Lastly, the prediction parameters are obtained based on the predictive control second optimization function. In the final stage, the corresponding predictive guidance track points are calculated for the parameters, which are obtained by the obstacle avoidance rules and weight functions of different obstacle environment types. Combining with the guidance [33,34] method, the AUV successfully avoids obstacles and gets to the position of an intended target by traveling over the predictive track. By using the obstacle avoidance method proposed, the AUV obstacle avoidance in an unknown underwater environment is successfully achieved. Experiments compared the work efficiency and task completion of AUV using APF and PSO algorithms, respectively. Lastly, it is proven via experimental data that, compared with the traditional search methods and conventional optimization algorithms mentioned above, the PGOA proposed has the clear advantage. Moreover, the simulation experiment fully demonstrates the adaptive characteristics of AUV in different environments with complex obstacles.

The rest of this paper is organized as follows. Section 2 introduces problem description and mathematical modeling. Section 3 presents environments where obstacles distribute differently. Section 4 develops AUV PGOA design. Section 5 gives a discussion of the results of experimental data in various situations. Lastly, conclusions are given in Section 6.

## 2. Problem Statement and Model Description

### 2.1. Problem Description

In unknown underwater environments, AUV may encounter complex terrain such as reefs, islands, trenches, and valleys, and the obstacles they meet are different in shape, complexity, and quantity. While ensuring the completion of underwater missions such as target search and intelligence reconnaissance, guaranteeing its own safety, it is necessary that AUV has the ability to avoid these complex obstacles in each situation. Therefore, it is important to analyze the factors that may affect cooperative searching for multi-AUVs.

1. Visual noise and threshold: Through the AUV-configured forward-looking sonar, the observation data is affected by Gaussian noise during the measurement of the observation target. In addition, AUV observation of the unknown environment through the forward-looking sonar is also limited by the sensor’s detection distance. As such, the sonar cannot observe and extract environmental features that are outside the view.

2. Movement limitation: The AUV’s own motion state will be affected by equipment such as thrusters and rudders. In an unknown underwater environment, the AUV will also be affected by unknown factors such as ocean currents and submarine topography. Therefore, the influence of the AUV motion restriction features on the track planning during the AUV navigation process need to be considered.

3. Obstacle types: When AUV performs tasks in an unknown underwater environment, it encounters a variety of obstacle types, such as simple convex obstacles, complex convex obstacles, and complex vortex obstacles. Therefore, a single obstacle avoidance algorithm cannot solve sudden problems. Therefore, different obstacle avoidance strategies and algorithms are developed in this paper for different obstacle types appearing in the AUV field of view, so as to achieve the optimal obstacle avoidance effect.

4. Obstacle avoidance: AUVs may operate in an unknown underwater environment, and it is inevitable for AUVs to encounter obstacles, which may threaten the AUV’s normal trajectory in the process of performing target searching tasks. Therefore, the AUV is expected to have the ability to avoid obstacles in a timely manner, to ensure that AUV can travel safely and reliably. Ultimately, the economic cost caused by equipment damage can be avoided and the searching task can be reliably completed.

### 2.2. AUV Movement Model

The AUV four-degrees-of-freedom constant-speed motion model $x_{t+1}=f(x_{t})$ was established to describe the form of AUV motion under water. In this paper, the updating of velocity and position follows the following formula.

In this paper, according to the standard AUV equation of motion [35], the updating of velocity and position follows the formula below.
(1)$$\left[\begin{array}{c} \dot{x}\\ \dot{y}\\ \dot{z} \end{array}\right]=R(\Theta )\left[\begin{array}{c} u\\ v\\ w \end{array}\right]$$
(2)$$R(\Theta )=\left[\begin{array}{ccc} \cos\psi \cos\theta & \cos\psi \sin\theta \sin\phi -\sin\psi \cos\phi & \sin\psi \sin\phi +\cos\psi \sin\theta \cos\phi \\ \sin\psi \cos\theta & \cos\psi \cos\phi +\sin\psi \sin\theta \sin\phi & \sin\psi \sin\theta \cos\phi -\cos\psi \sin\phi \\ -\sin\theta & \cos\theta \sin\phi & \cos\theta \cos\phi \end{array}\right]$$

Because the roll movement is uncontrollable for AUV and the simulation environment of the obstacle avoidance algorithm is built on the 2-D environment, defining ϕ,θ = 0. Considering that the AUV’s additional hydrodynamic resistances in the horizontal and vertical direction are greater than those in the longitudinal one, when the speed over grand (SOG) exceeds 1 knot, the propulsive efficiency of the auxiliary thrusters is very low. Therefore, when the AUV is navigating at a normal speed, its auxiliary thrusters are idle. Usually, we take w = 0, v = 0, and then Equations (1) and (2) are simplified as [36]:(3)$$\left\{ \begin{array}{l} u=v_{c}\\ \overset{\cdot }{x}=u\cos(\psi )\\ \dot{y}=u\sin(\psi )\\ \overset{\cdot }{\psi }=r\\ x(t+1)=x(t)+\overset{\cdot }{x}\\ y(t+1)=y(t)+\overset{\cdot }{y} \end{array}\right.$$
where (x(t),y(t)) represents the positioning information that is related to the time variable t, $\left(\overset{\cdot }{x},\overset{\cdot }{y}\right)$ represents the velocity vector of the AUV in the global coordinate system, $\upsilon_{c}$ is a normal constant, and ψ represents the angle between the AUV direction and the direction of the global coordinate system axis, and *r* is a heading velocity variable in this case. AUV is affected by its own equipment, and its speed $\upsilon_{c}$ and corner ψ are limited.

### 2.3. Forward-Looking Sonar Model

Based on the unknown underwater environment, the real multi-beam active forward-looking sonar data is simulated through a mathematical model of forward-looking sonar [37] in this paper. According to the common multi-beam sonar background, the Seabeat6012 sonar is selected as the forward-looking sonar of the AUV. The Seabeat6012 sonar has a visible range *R* of 150 m, a horizontal opening angle α of 120°, a vertical opening angle β of 15°, and an operating frequency of 2 kHz. To obtain the target data, the sonar model performs the array statistics $d\in R^{80x3}$ of the range of the sonar opening angle. Then based on the elements filled in the matrix, it judges whether there is a target in a certain position in the visible range. The FLS can be roughly represented, as shown in Figure 1.

The mathematical model for FLS is established, which can describe the constraint relationship between the target and the FLS. Then the data information of the detected object is obtained. The model can be given by the equation below.
(4)$$\{\begin{array}{c} \frac{\left|y_{t}\right|}{\sqrt{x_{t}^{2}+y_{t}^{2}}}\leq \sin\frac{\alpha }{2}\\ \sqrt{x_{t}^{2}+y_{t}^{2}+z_{t}^{2}}\leq R\\ \frac{\left|z_{t}\right|}{\sqrt{x_{t}^{2}+y_{t}^{2}}}\leq \sin\frac{\beta }{2} \end{array}$$
where $\left(x_{t},y_{t},z_{t}\right)$ can be expressed by the equation below.
(5)$$\{\begin{array}{c} x_{t}=x-x_{0}\\ y_{t}=y-y_{0}\\ z_{t}=z-z_{0} \end{array}$$
where (x,y,z) is the coordinate of the target in the hull coordinate system $\left(Ox_{0}y_{0}z_{0}\right)$. $\left(x_{0},y_{0},z_{0}\right)$ is the body coordinate of the AUV sonar. $\left(x_{t},y_{t},z_{t}\right)$ denotes the relative positional relationship between the target and the AUV. By judging the position and angle relationship of $\left(x_{t},y_{t},z_{t}\right)$, this model is determined whether the target is in the sonar field.

Because the forward-looking sonar equipped on the AUV can be easily affected by the medium of water or other external factors during the data collection process, such as data interference, the measurement of the environmental characteristics is likely to be affected. Therefore, the description of the sonar is given in Equation (6) [38].
(6)$$y_{x-q}=\left\{ \begin{array}{l} none|x-q|>L\\ none|x-q|\begin{array}{cc} in & \mathrm{Obstacles} \end{array}\\ h(x,q)+d(x,q)\zeta \begin{array}{cc} & |x-q|<L \end{array} \end{array}\right.$$
where $y_{x-q}$ denotes the FLS measurement from an AUV at position x to a sensing point at position q. none indicates that the environmental feature data does not exist. L is the visual threshold. h(x,q) is the sensor model in the noise free case, d(x,q) is the distance between x and q, and ζ is standard Gaussian noise.

The above description indicates that, when the relative position of the FLS and the detected object is outside the sight range, or there is an obstacle between the sonar and the detected object, obtaining feedback of characteristic information is unavailable. If there is characteristic information in the sight range, the observed data indicates the distance increases, and the measured disturbance increases, accordingly.

## 3. Classification of Obstacles and Solutions

### 3.1. Type of Obstacles

For a wide expanse of unknown waters, when AUVs are dispatched to perform underwater operations, they often encounter unpredictable harsh environments with obstacles everywhere, which affects the normal movement of AUVs. For example, Figure 2 lists several typical environments of obstacle distributions: environments of convex obstacles, vortex obstacles, and dense convex obstacles.

(1) Convex Obstacles

In this paper, the sporadic distribution of a small number of convex obstacles is defined as a simple convex obstacle environment type in an unknown environment in Figure 2b. In this situation, AUV could easily avoid collisions and plan a safe and reasonable trajectory to complete the mission successfully.

(2) No-Convex Obstacles

In some cases, AUV can encounter some non-convex obstacles, such as concave structures or helical shape. In such a case, a simple obstacle avoidance algorithm is not enough for AUVs escaping from obstacles in Figure 2a. When an AUV is stuck in such an obstacle, a corresponding escape algorithm needs to be developed to escape the vortex obstacles.

(3) Dense Convex Obstacles 

Part 2.1(3) describes the types of obstacle structures. In reality, there is also an environment of densely distributed convex obstacles in Figure 2c, which demands higher obstacle avoidance abilities for AUV. Therefore, in intensive obstacle environments, a safe trajectory needs to be planned for AUV to safely navigate through the densely distributed obstacles region.

Figure 2 shows specific characteristics and distributes for each type obstacle in unknown environments. The red spot is a virtual target point, which indicates that AUV has completed its task. To verify the effectiveness of the algorithm, an unknown environment with an area of 2000 m × 2000 m is designed, which is divided into 400 m × 400 m task sub-areas, which supposes that the obstacle information is unknown for the AUV. For the three types of obstacles mentioned above, different obstacle avoidance methods are designed for AUV to complete its task and reach the desired target point safely. The following solutions are developed to three situations.

### 3.2. Obstacle Detection Principles

AUV uses FLS to realize the collision avoidance function. A real-time obstacle avoidance strategy is proposed in this paper. All obstacles in the environment are unknown, and their shapes and positions are randomly generated. The obstacle boundary is generated based on the detected information of the FLS. Considering that the AUV pitch angle rarely changes, the multi-beams sector on the horizon in the body coordinates is used. The purple lines are the sonar beams on the horizontal plane in the AUV body coordinates. The gray part is the obstacle, and the blue boundary is the obstacle contour curve (Figure 3).

### 3.3. Obstacle Condition Classification

To improve the effectiveness and safety of the AUV in obstacle avoidance when performing underwater tasks, the obstacles are divided into four conditions according to the positions of obstacles relative to AUV: bounded obstacles, left bounded obstacles, right bounded obstacles, and left and right edge unbounded obstacles. When the obstacle enters the FLS detection range, the sonar will classify the obstacle based on detected data by emitting 80 beams. The detection zone is a fan-shaped range of 80 m, as shown in Figure 4. In addition, k and l is the left boundary and right boundary detected by FLS, respectively, *i*, *j* are the serial numbers of beams, δ, ζ is the arbitrary nature number [39].

(1) If the boundary of the obstacle is in the beam range of the FLS, the current obstacle is considered as the bounded obstacle (BO).
$$\exists k,l,\delta ,\varsigma \in Z,k<l,i,j\in N^{+},i\in [k,l],j\in [k-\varsigma ,k-1]\cup [l+1,l+\delta ],st.0<S_{i}\leq L_{e},st.0<S_{j}\leq L_{e}$$

(2) If the right boundary of the obstacle is outside the sonar beam range, and the left boundary is in the range, the detected obstacle is called the left bounded obstacle (LB).
$$\exists k,\delta \in Z,i\in [1,k],j\in [k,k+\delta ],st.0<S_{j}\leq L_{e},S_{i}=0$$

(3) If the left boundary of the obstacle is outside the detection range of the sonar, and the right boundary is in the range, the obstacle is defined as the right bounded obstacle (RBO).
$$\exists l,\varsigma \in Z,i\in [l-\varsigma ,l],j\in [l,80],st.0<S_{i}\leq L_{e},S_{j}=0$$

(4) If both sides of the obstacle are outside the detection range of the sonar, the obstacle is defined as an unbounded obstacle (UBO).
$$\exists i,j\in [1,80],st.0<S_{i}\leq L_{e},st.0<S_{j}\leq L_{e}$$

### 3.4. Obstacle Avoidance Boundary Data Processing

First, the obstacle information in the sight range is obtained using FLS of AUV. The detected data is stored in the matrix $\beta \in R^{80\times 3}$, and only the horizontal plane of the sonar opening angle is used in this paper, so just the second column of data of γ∈β is needed, with each representing the object distance and angle information detected by the sonar beam. If there are elements in γ equal to zero, it means that the sonar beams do not detect any objects. Group data is considered an obstacle. For example, the detected data can be divided into two groups in Figure 5.

Data in γ is grouped based on Formulas (10) and (11). First, the proper sonar beam spacing is selected based on the FLS type. Then based on Equation (10), the obstacle information is judged whether it is a continuous sonar beam data, and whether the 2-norm of the obstacle boundary point satisfies the beam spacing condition. The obstacle data can be grouped by iterating boundary data.
(7)$$\left||\bar{S_{i}S_{i-1}}||<d_{b},S_{i}\cdot S_{i-1}\neq 0,i\in [2,80\right]$$
(8)$$d_{b}=\frac{\lambda_{t}(l_{e}\cdot \varphi_{s})}{n-1}\begin{array}{cc} , & (n=80 \end{array},\lambda_{t}\in [1,4])$$
where $d_{b}$ is the beam spacing. $S_{i}$ is the obstacle point detected. $\lambda_{t}$ is the screening factor, and $l_{e}$ is the detection range of the sonar.

FLS can output a data structure for the detected obstacles, where the obstacle bounder data are stored in the same array. In reality, the obstacle shapes are always irregular, so the output data from the sonar cannot be directly used to avoid obstacles. In this paper, the convex hull algorithm combined with the simplified Bezier interpolation algorithm is used to transform the obstacle outline into a regular shape, and the data is smoothed. It not only improves the accuracy of obstacle avoidance, but also avoids the adverse effects of complex obstacles. In Figure 6, when the sonar detects a bounded obstacle, to improve the accuracy of the obstacle avoidance efficiency, it is necessary to simplify and smooth the boundary of the obstacle. By using the horizontal plane layer data $\beta \in R_{2}{}^{80\times 2}$, coordinate transformation is performed on the distance and angle data from the sonar beam point, since the input data streams into the convex algorithm. When the data is simplified, it can be smoothed using Bezier interpolation. To improve the algorithm speed, this paper selects 80 variables to segment the smooth data. The specific implementation process is as follows.

Step 1: The data output from the sonar beam line is solved as an obstacle boundary point.
(9)$$\left\{ \begin{array}{l} O_{ix}=a_{uvx}+d_{i}\cos(\psi -\theta_{i})\\ O_{iy}=a_{uvy}+d_{i}\sin(\psi -\theta_{i}) \end{array}\right.\begin{array}{cc} , & \end{array}(\theta_{i}\in [\psi ,\psi +\pi /3],i\in [k,l],k,l\in N^{+})$$
(10)$$\left\{ \begin{array}{l} O_{ix}=a_{uvx}+d_{i}\cos(\psi +\theta_{i})\\ O_{iy}=a_{uvy}+d_{i}\sin(\psi +\theta_{i}) \end{array}\right.\begin{array}{cc} , & \end{array}(\theta_{i}\in [\psi -\pi /3,\psi ),i\in [k,l],k,l\in N^{+})$$
where ψ is the current heading of the AUV. $d_{i}$ denotes the distance of the i-th sonar beam output. $\theta_{i}$ is the angle of the i-th beam output, and $a_{ix},a_{iy}$ is the position of the current AUV.

Step 2: For the set X of the obstacle boundary points calculated by the solution, a convex hull set, which includes all the points in the X set, needs to be found to replace the X set. Then we select the leftmost and lowest point in the set X as the origin of the polar coordinates. Then we sequence all the points in the set, based on the principle that their distance from the polar origin is short to long and the polar angle ranges from small to large. Subsequently, the ray is selected, which is generated by the second point after sorting and the polar coordinates. Then the vector angle from small to large is found by using the vector cross method. The convex hull is found, according to the distance from small to large, if the angles are the same.

Step 3: The convex hull set, calculated in step 2, represents a simplified set of contour points of the obstacle that are used as boundary points of the Bessel interpolation to generate the smooth obstacle boundary points. The specific calculation is shown below.
(11)$$B(t)=\sum_{i=0}^{n}C_{i}^{n}P_{i}{\left(1-t\right)}^{n-i}t^{i}\begin{array}{ccc} & & t\in [0,1] \end{array}$$
where $P_{i}$ is the interpolation point $P_{i}=(x_{i},y_{i})$, and the cubic Bessel interpolation function is used in this paper.
(12)$$\begin{array}{l} B_{3}(t)=\sum_{i=0}^{3}C_{i}^{3}P_{i}{\left(1-t\right)}^{3-i}t^{i}\\ \begin{array}{ccc} & & ={\left(1-t\right)}^{3} \end{array}P_{0}+3{\left(1-t\right)}^{2}tP_{1}+3(1-t)t^{2}P_{2}+t^{3}P_{3}\begin{array}{cc} & t\in [0,1] \end{array} \end{array}$$

Step 4: The last processed boundary points are sorted from left to right in the order of the sonar beam. The final result is shown in the following figure. The red line is the contour line processed by the convex algorithm, and the blue line is the final processing result.

## 4. Predictive Guidance Obstacle Avoidance Algorithm Design

### 4.1. AUV Maximum Obstacle Avoidance Turning Radius

When AUV navigates at 2 m/s in an underwater environment without ocean currents, and the rudder angle is set at a maximum steering angle of 35°, the minimum turning radius is about five times the length of the AUV. It takes approximately 3.5 seconds for the rudder angle of the AUV to vary from 0° to 35°. If the time delay of the steering angle transition is considered, the trajectory deviation distance is 1–1.5 m, which is small compared to the turning radius. To simplify the problem that the deviation distance is neglected. In other words, the trajectory deviation rotation is replaced by an arc with a certain radius.

In Figure 7, Di is the best obstacle avoidance point detected by the FLS. Based on the current AUV state information and the obstacle data collected by the FLS, the angle and distance information ($\alpha_{i}$,$\rho_{i}$) relative to the AUV of the current obstacle point can be known, where $\alpha_{i}$ is the angle between the AUV heading and the obstacle point detected by the sonar, and $\rho_{i}$ is the distance between the obstacle point and the current AUV. $R_{i}$ denotes the maximum turning radius relative to the current AUV. *o* is the center of the circle where the maximum conversion radius is located. ob is the mid-perpendicular line of Ac, and $D_{s}$ is the safety distance. The specific definition is as follows:

$∠dAD_{i}=i\alpha_{i}$,$∠dAc=\beta_{i}$,$\bar{AD_{i}}=\rho_{i}$,$\bar{D_{i}c}=D_{s}$,$\bar{Ao}=R_{i}$.

The maximum turning radius is expressed as follows [39].
(13)$$\frac{\bar{AD_{i}}}{\sin(∠D_{i}cA)}=\frac{\bar{D_{i}c}}{\sin(∠D_{i}Ac)}$$
(14)$$\left\{ \begin{array}{l} ∠D_{i}Ac=\beta_{i}-\alpha_{i}\\ ∠D_{i}cA=\pi -(\pi /2-\beta_{i})=\pi /2+\beta_{i} \end{array}\right.$$

Based on Equations (16) and (17):(15)$$\left\{ \begin{array}{l} \alpha_{i}=\varphi_{s}(i-0.5n+0.5)/n\\ \beta_{i}=\frac{D_{s}}{\rho_{i}}\sec\alpha_{i}+\tan\alpha_{i} \end{array}\right.$$
(16)$$\frac{\bar{AD_{i}}}{\sin(∠D_{i}oA)}=\frac{\bar{oA}}{\sin(∠oD_{i}A)}$$

The maximum turning radius obtained by Equations (18) and (19) is as follows:(17)$$R_{i}=\rho_{i}[\cos\alpha_{i}-\sin\alpha_{i}/\tan(2\beta_{i})]$$

If AUV avoids obstacles by rounding the right edge of the obstacle, then $R_{i}$ denotes the maximum turning radius of the obstacle point detected by the i-th sonar beam. The expected maximum turning radius is expressed as follows.
(18)$$R_{\max}=\min\{R_{i}|i=41,42,\ldots ,80\}$$

Otherwise, when the AUV avoids the obstacle around the left edge of the obstacle, the expected maximum turning radius is expressed as follows.
(19)$$R_{\max}=\min\{R_{i}|i=1,2,\ldots ,40\}$$

### 4.2. AUV Obstacle Avoidance Rules

Safe obstacle avoidance distance and the emergency obstacle avoidance distance is designed in the AUV obstacle avoidance process. Therefore, the following rules are formulated to deal with the two existing situations.

Rule 1: When there is a safe obstacle avoidance distance

a. If it is a left bounded obstacle that meets the conditions for safe obstacle avoidance, the AUV turns to the left side of the obstacle to avoid it.

b. If it is a right bounded obstacle that meets the conditions for safe obstacle avoidance, the AUV turns to the right side of the obstacle to avoid it.

c. If it is a bounded obstacle or an unbounded obstacle that meets the safe obstacle avoidance distance, the AUV turns in the direction close to the virtual target to avoid the obstacle.

Rule 2: When an emergency obstacle avoidance situation occurs, a corresponding rule needs to be established to ensure the safety of the AUV. To ensure absolute safety, in practice, it is necessary to immediately turn off the propeller and initiate reverse propulsion in order to offset the forward speed caused by inertia. However, this extreme situation is not considered in this article.

a. If there is an emergency obstacle avoidance setting where a bounded obstacle or an unbounded obstacle exists, the current AUV heading is taken as the dividing line to estimate the boundary point of the obstacle that is closer to the virtual target. Then based on the boundary point data generated, we use the method mentioned above to calculate the maximum turning radius. Lastly, the minimum turning radius that meets the safe obstacle avoidance distance is selected.
(20)$$\left\{ \begin{array}{l} \lambda_{r}>\lambda_{l}\begin{array}{cc} & \mathrm{choose}\begin{array}{cc} & \mathrm{left} \end{array} \end{array}\\ \lambda_{r}\leq \lambda_{l}\begin{array}{cc} & \mathrm{choose}\begin{array}{cc} & \mathrm{right} \end{array} \end{array} \end{array}\right.$$
(21)$$\left\{ \begin{array}{l} D_{v}\sin(\lambda_{r})-2\mathrm{sgn}(R_{\min})R_{\min}^{\delta }>D_{v}\sin(\lambda_{l}),\begin{array}{cc} \mathrm{turn} & left \end{array}\\ else,\begin{array}{cc} \mathrm{turn} & right \end{array} \end{array}\right.$$
where $D_{v}$ is the distance of the AUV from the virtual target. $R_{\min}$ is the minimum radius of the radius, which meets a maximum turning radius. $\lambda_{r},\lambda_{l}$ is the right and left angle of the obstacle avoidance path away from the target, respectively. δ is the set constant influence factor.

b. If there is an emergency obstacle avoidance and the bounded obstacle is in the field of view of the sonar, in order to perform a safe and most energy-efficient obstacle avoidance, the AUV sails toward the bounded side while calculating the turning radius.

The obstacle avoidance rule flow chart is shown in Figure 8.

### 4.3. Constructing the Weighting Function of the Obstacle Avoidance Algorithm

#### 4.3.1. Weight Function for Avoiding Influencing Factors

According to the principle of the predictive control model [40], the proposed predictive step size is M. To ensure the safety of the AUV, the optimal obstacle avoidance trajectory points in the locally known area of the AUV are obtained. Five factors affecting obstacle avoidance are considered: safety, rate of change of AUV yaw angle turning radius, obstacle avoidance path, and obstacle avoidance area mode.

• The Weight Function of Safe Distance

To improve the safety of AUV obstacle avoidance, the relationship between the reference obstacle point of the AUV selection and the current AUV distance as well as the safety distance must be considered. The relationships are shown below.
(22)$$\begin{array}{l} d_{s}(t+m)=||A_{uv}(x,y)-O_{i}(x,y)||\begin{array}{cc} , & i=k,\ldots ,l \end{array}\\ \gamma_{s}(t+m)=\frac{\left|d_{s}(t+m)-D_{s}\right|}{D_{s}}\\ f_{s,value}=k_{1}\gamma_{s}(t+m) \end{array}$$
where $d_{s}$ is the relative distance between the AUV and the boundary point of the obstacle is detected by the sonar. $D_{s}$ is the safety distance. $\gamma_{s}$ is the safety threshold, and $k_{1}$ is the weight coefficient.

Clearly, the weight function can reduce the impact of this problem on the obstacle avoidance algorithm. The risk of obstacle avoidance failure due to a single collision avoidance condition can be avoided.

• Weight Function for the Rate of Change of the AUV Yaw Angle

To prevent the shaking movement of the yaw angle during the control process, the influence of the rate of change of the AUV yaw angle is introduced to achieve the smooth control of the AUV turning heading. The weight function is defined below.
(23)$$f_{z,value}=k_{2}(\psi_{t+m+1}-\psi_{t+m})$$
where $k_{2}$ is the weight coefficient, and $\psi_{t+m}$ is the AUV heading at a certain moment.

• Weight Function of the Turning Radius

The predictive track points are used for guidance control to achieve local obstacle avoidance. According to the method of calculating the turning radius proposed above, the maximum turning radius cannot be directly used as the guidance parameter, because the turning radius is only one factor that affects the obstacle avoidance performance. The specific weight function is shown below.
(24)$$\begin{array}{l} \alpha_{t+m}=a\tan2(A_{uv,t+m}(x,y)-O(x_{o},y_{o}))\\ \beta_{t+m}=\frac{b^{2}+a^{2}-o^{2}}{2ab}\\ \gamma =\left\{ \begin{array}{l} \alpha_{t+m}\begin{array}{cc} , & A_{uv,t,x}-S_{x_{o}}\geq 0,A_{uv,t+m,y}\leq S_{y_{o}} \end{array}\\ \begin{array}{ccc} \beta_{t+m}+\pi /2, & & A_{uv,t+m,x}-S_{x_{o}}<0,A_{uv,t,y}\leq S_{y_{o}} \end{array}\\ -\alpha_{t+m}\begin{array}{cc} , & A_{uv,t,x}-S_{x_{o}}\geq 0,A_{uv,t+n,y}\geq S_{y_{o}} \end{array}\\ \begin{array}{ccc} -\beta_{t+m}-\pi /2, & & A_{uv,t+m,x}-S_{x_{o}}\leq 0,A_{uv,t+m,y}\geq S_{y_{o}} \end{array} \end{array}\right.\\ A_{uv,t+m,x}=O_{x_{o}}+R\cos(\gamma )\\ A_{uv,t+m,y}=O_{y_{o}}+R\sin(\gamma ) \end{array}$$
(25)$$f_{l,value}=k_{3}||A_{uv}(x_{t+m},y_{t+m})-T_{\arg et}(x,y)|{|}_{2}$$
where γ is the angle between the position of the maximum turning radius where the AUV is located and the center of the circle is shown as the polar coordinate origin. R is the turning radius. $S_{x_{o}},S_{y_{o}}$ is the coordinate of the obstacle point where the maximum turning radius is located, and $A_{uv,t+m}(x,y)$ is the real time position of the AUV obstacle avoidance track. $T_{\mathrm{arget}}(x,y)$ is the virtual target point, and $k_{3}$ is the weight coefficient.

• Weight Function of the Obstacle Avoidance Path

To ensure that the AUV is able to reach the desired virtual target point by the shortest travel path within the predictable range, the weight function of the obstacle avoidance path can be defined by the equation below.
(26)$$f_{d,value}=k_{4}{\int_{t=0.1}^{t=T}\left||A_{uv,t+m}(x,y)+A_{uv,t+m+1}(x,y)|\right|}_{2}dt$$
where T denotes the time it takes to reach the desired target. $A_{uv,t+m}(x,y)$ is the AUV position information at time m, and $k_{4}$ is the weight coefficient.

• Weight Function of the Obstacle Avoidance Area

To improve the reliability of obstacle avoidance, the fan-shaped field of view detected by FLS is generally divided into three parts: non-avoidance areas, safe obstacle avoidance areas, and emergency obstacle avoidance areas. Therefore, it is necessary to judge which area the obstacle detected belongs to, and adopt different obstacle avoidance strategies. Therefore, the weight function of the obstacle avoidance area is defined below.
(27)$$f_{a,value}=k_{5}\cdot \left\{ \begin{array}{l} 1\begin{array}{cc} , & d_{i}\in (l_{s},l_{e}] \end{array}\\ 0\begin{array}{cc} , & \end{array}d_{i}\in [l_{d},l_{s})\\ -1\begin{array}{cc} & d_{i}\in [D_{s},l_{d}) \end{array} \end{array}\right.$$
where $d_{i}$ is the distance between the obstacle boundary point and the current position of the AUV. $l_{e},l_{s},l_{d},D_{s}$ is the longest distance that the sonar can detect, the maximum distance of the safe obstacle avoidance area in the field of view of sonar, the maximum range of emergency obstacle avoidance area, and safety distance, respectively. $k_{5}$ is the weight coefficient.

#### 4.3.2. Conditional Constraints of Weight Function

Due to the impact of physical structure characteristics of the AUV itself, and the obstacle avoidance environment, several constraints must be met to achieve the purpose of collision avoidance successfully.

a. According to the structural characteristics of the physical design of the AUV, Maximum turning angular velocity meets: $-35^{\circ }\leq \omega_{t}\leq 35^{\circ }$;

b. To ensure the safety of the AUV, the safety distance is set to meet the conditions: $5l_{o}\leq D_{s}\leq 6l_{o}$, where $l_{o}$ is the length of the AUV,

c. The maximum turning radius meets the conditions: $R\in [R_{\max},R_{\min}]$.

#### 4.3.3. Conditional Constraints of Weight Function

Combining weight functions, model parameters, and the obstacle avoidance principle, the secondary optimization function based on predictive control is obtained below.
(28)$$\min\sum_{m=0.1}^{N-1}||f_{s,value}|{|}^{2}+||f_{z,value}|{|}^{2}+||f_{l,value}|{|}^{2}+||f_{d,value}|{|}^{2}+||f_{a,value}|{|}^{2}$$

Since Equation (27) introduces non-contiguous Boolean variables as the weight function, the objective function (28) is made non-convex. The corresponding problem becomes the mixed integer nonlinear programming (MINLP) problem, but, so far, this type of problem has not found a unanimous and mature solution [41]. However, it can be seen from Equation (27) that Boolean variables introduced are only used for the evaluation of the weight function, but not for state variables or control variables that need to be optimized. Therefore, it is still essentially nonlinear programs (NLP) problem with constraints. Furthermore, there are many excellent algorithms for solving NLP problems [42]. However, since the weight function (27) is non-convex, such NLP problems can only obtain a local optimal solution. To weaken the influence of Boolean variables, the Boolean variables of the weight function (27) are relaxed into a contiguous space, by using the hyperbolic tangent function in the Sigmoid function as follows.
(29)$$f^{\prime }{}_{a,value}=k_{5}t\mathrm{anh}(\mu *d_{i}-80)=k_{5}\frac{1-\exp(-\mu d_{i}-80)}{1+\exp(-\mu d_{i}-80)}$$
where μ is the slope of the Sigmoid function. $d_{i}$ is the continuous variable of the feasible domain.

The weight function before and after relaxation is shown in Figure 9. In addition, it has a good approximation and retains the property of the original weight function. The continuous convex problem is obtained after the relaxation of Boolean variables, and the predicted trajectory point of the AUV local obstacle avoidance can be obtained using the convex NLP problem. Thus, we can use the existing Sequential Quadratic Programming (SQP) method to solve the problem [43,44].

### 4.4. Overview of AUV Obstacle Avoidance Algorithms

The predictive guidance control obstacle avoidance algorithm is a real-time collision avoidance algorithm that ensures the safety of the AUV when the AUV performs tasks. The details are as follows:

(1) Always moving toward the virtual target point, which is the direction of the AUV’s minimum cost. When no obstacles or many obstacles detected by the FLS configured on the AUV are outside the obstacle avoidance distance, the AUV needs to move in the direction of the target point where the greatest profit is obtained.

(2) Maintaining a safe distance: When the obstacle detected by the FLS meets the obstacle avoidance distance, the safety distance must be considered in the AUV collision avoidance. The length of the AUV is 5 m, and the safety distance is generally four times longer than the AUV’s length. When selecting the obstacle avoidance guidance point, it is necessary to judge whether the safety distance meets the safe distance of the obstacle avoidance.

(3) Simple convex obstacles: When the AUV keeps the current heading in a certain area, and a single convex obstacle is detected by the FLS, the appropriate track points are chosen, according to the obstacle avoidance rule. This uses the dynamic guidance algorithm to adjust the heading in real time, and, at the same time, control the appropriate speed to achieve safe obstacle avoidance.

(4) Vortex obstacles: In the current heading path of the AUV, there may be a complex vortex obstacle. Since the obstacle appears only partly in the detectable area of the sonar and it is too hard to identify, the AUV may enter inside the concave obstacle. Therefore, it is necessary to establish obstacle avoidance rules for concave obstacles, and, based on the obstacle avoidance rules mentioned above, the effective real-time algorithm can be achieved between the two obstacle avoidance guidance algorithms along the vortex obstacle wall for the AUV to drive safely and, lastly, escape the vortex obstacles.

(5) Dense convex obstacles: When more than two obstacles were detected by the forward-looking sonar of the AUV, the above two methods of obstacle avoidance are not fully competent. For the AUV to sail safely in the multiple convex obstacle environments, we combed the predictive guidance obstacle avoidance algorithm and obstacle avoidance rules based on different classifications of multiple obstacle environments. The specific algorithm for such a setting is described in Section 4.5.3.

### 4.5. Different Obstacle Avoidance Algorithm Designing Various Types of Obstacles

To ensure the AUV complete its task safely, it is necessary to design reasonable obstacle avoidance algorithms for different types of obstacle environments so that the AUV can adapt to the harsh obstacle environments when sailing in an unknown underwater environment with obstacles. Therefore, by using the predictive guidance control obstacle avoidance algorithm, we can obtain the shortest and smoothest obstacle avoidance path, so that the AUV can successfully achieve collision avoidance with less rotation and rudder angle correction. Obstacle avoidance track points are predicted based on the obstacles detected by sonar. Then, the AUV follows the obstacle avoidance track points by linear guidance and arc guidance (the radius is the turning radius) to achieve a safe obstacle avoidance path.

#### 4.5.1. Obstacle Avoidance Algorithm Designing for Simple Convex Obstacles

The AUV uses the obstacle avoidance rule mentioned above, and the obstacle avoidance weight function is used to calculate the appropriate trajectory point to get the best track point and bypass the obstacle. Taking the right bounded obstacle in Figure 10 as an example, the predictive guidance avoidance algorithm is designed as follows.

1. By processing the boundary data of obstacles that has been detected by FLS, the number of obstacles and their relative positions to the AUV can be obtained.

2. The relation between the obstacle points detected by sonar beam and the current AUV is calculated by using the optimal parameters obtained by the weight function and the coordinate information of all points are calculated by using the method mentioned above.
(30)$$P_{j,i,\min}=\min(D=[(O_{ix},O_{iy}),i=1,\ldots ,m])\begin{array}{cc} , & m\leq n \end{array}$$
(31)$$P_{j,i,\max}=\max(D=[(O_{ix},O_{iy}),i=1,\ldots ,m])\begin{array}{cc} , & m\leq n \end{array}$$
(32)$$Q_{\min j,i}=\{(P_{j,i,\min\_x},P_{j,i,\min\_y}),d_{j,i,\min},\theta_{j,i}\}\begin{array}{cc} , & \theta_{j,i}\in [\psi -\pi /3,\psi +\pi /3] \end{array}$$
(33)$$Q_{\max j,i}=\{(P_{j,i,\max\_x},P_{j,i,\max\_y}),d_{j,i,\max},\alpha_{j,i},\}\begin{array}{cc} , & \theta_{j,i}\in [\psi -\pi /3,\psi +\pi /3] \end{array}$$
where $P_{j,i,\min}$ is the beam spot with the shortest distance from the AUV in all detected beam points. D is the set of all beam points. $Q_{\min j,i}$ is the set of all information of the shortest distance point. $Q_{\max j,i}$ is the set of all information of the longest distance point, and ψ is the current AUV heading.

3. Judging whether the current point is on the left or right side of the sonar center line is demarcated by the AUV heading. If it is on the left side and the result of the classification using step 1 is applied, for example, j=1, then it indicates that it is a succession of single obstacles, and judges the relationship between $d_{j,i,\min}$ and the general obstacle avoidance distance $l_{s}$ as well as the warning obstacle avoidance distance $l_{d}$.
(34)$$\eta_{\mathrm{obs}}=\frac{abs(\theta_{j,i}-\alpha_{j,i})}{\pi *2/3}$$
(35)$$\left\{ \begin{array}{l} d_{j,i,\min}\in [l_{d}\begin{array}{cc} ,l_{s}], & SOA \end{array}\\ d_{j,i,\min}\in [D_{s},l_{d}]\begin{array}{cc} , & EOA \end{array} \end{array}\right.$$
(36)$$\left\{ \begin{array}{l} \eta_{\mathrm{obs}}\in [0,0.5]\begin{array}{cc} , & Hobs \end{array}\\ \eta_{\mathrm{obs}}\in (0.5,0.8]\begin{array}{cc} , & OHobs \end{array}\\ \eta_{\mathrm{obs}}\in (0.8,1]\begin{array}{cc} , & Eobs \end{array} \end{array}\right.$$
where $\eta_{\mathrm{obs}}$ is the proportion of the obstacle take in the sonar’s field of view occupied by obstacles. SOA is the safe obstacle avoidance range. EOA is the emergency obstacle avoidance range. Hobs is the primary sonar sight range proportion. OHobs is the intermediate sonar sight range proportion. Eobs is the emergency sonar sight range proportion.

4. If $P_{j,i,\min}\in [S_{i},i=1,\ldots ,40]$ and it meets the SOA and Hobs conditions, then the predictive obstacle avoidance guidance point is derived as follows and, after calculation, it goes to step 6 for execution.
(37)$$\delta =\frac{abs(P_{j,i,\max}(x,y)-P_{\psi }(x,y))}{abs(P_{j,i,\max}(x,y)-S_{80}(x,y))}$$
(38)$$P_{j,i,guide}=S_{80}(x,y)-\delta \cdot (1-abs(S_{80}(x,y)-\frac{S_{80}(x,y)}{P_{\psi }(x,y)}))\cdot S_{80}(x,y)$$
(39)$$P_{j,i,guide}=\varepsilon \cdot abs(q_{j,i}(x,y)-\frac{q_{j,i}(x,y)}{S_{80}(x,y)})+q_{j,i}(x,y)$$

If $P_{j,i,\min}\in [S_{i},i=41,\ldots ,80]$, and it meets the SOA and Hobs conditions, then the predictive obstacle avoidance guidance point is as follows and, after calculation, it goes to 6 for execution.
(40)$$q_{j,i}=\frac{\left(P_{j,i,\max}(x,y)+S_{80}(x,y)\right)}{2}$$
(41)$$\varepsilon =\frac{abs(q_{j,i}(x,y)-P_{j,i,\min}(x,y))}{abs(P_{j,i,\min}(x,y)-S_{80}(x,y))}$$
(42)$$P_{j,i,guide}=\varepsilon \cdot (1+abs(q_{j,i}(x,y)-\frac{q_{j,i}(x,y)}{S_{80}(x,y)}))+q_{j,i}(x,y)$$
where δ,ε are the remaining proportion of obstacles occupying the sight range of the sonar.

5. If $P_{j,i,\min}\in [S_{i},i=1,\ldots ,40]$ and it meets the SOA and Eobs conditions, then the predictive obstacle avoidance guidance point is $P_{j,i,guide}=S_{80}(x,y)$, go to 7 for execution.

If $P_{j,i,\min}\in [S_{i},i=1,\ldots ,40]$ and it meets the SOA and OHobs conditions, then the predictive obstacle avoidance guidance point is $P_{j,i,guide}=S_{80}(x,y)$. Go to 7 for execution.

If $P_{j,i,\min}\in [S_{i},i=41,\ldots ,80]$ and it meets the SOA and Eobs conditions, then the predictive obstacle avoidance guidance point is $P_{j,i,guide}=S_{1}(x,y)$, go to 7 for execution.

If $P_{j,i,\min}\in [S_{i},i=41,\ldots ,80]$ and it meets the SOA and OHobs conditions, then the predictive obstacle avoidance guidance point is $P_{j,i,guide}=S_{1}(x,y)$, go to 7 for execution.

6. Based on the obstacle avoidance track point obtained above, we can perform the following obstacle avoidance guidance algorithm to correct the position and heading of the AUV to sail a safe obstacle avoidance path.
(43)$$\begin{array}{l} \beta_{i}=a\tan2(P_{j,i,guide}(x,y)-A_{uv}(x,y))\\ \delta (t)=\beta_{i}-a\tan2(P_{j,i,guide}(x,y)-A_{uv}(x(t),y(t)))\\ d(t)=||P_{j,i,guide}(x,y)-A_{uv}(x(t),y(t))|{|}_{2}\\ \varepsilon (t)=d(t)\cdot \sin(\delta (t))\\ \psi_{d}=\beta_{i}-\alpha (t) \end{array}$$
where the selection of α(t) has certain rules as follows: When the current position of the AUV is far from the desired path and ε(t)>Δ, the front-looking vector has no intersection with the path. Then α(t) is selected as an angle perpendicular to the path direction, and it is π/2. If the current position of the AUV is closer to the desired path, then α(t)=asin(ε(t)/Δ).
(44)$$\alpha (t)=\left\{ \begin{array}{l} a\sin(\varepsilon (t)/\Delta ),|\varepsilon (t)|\leq \Delta \\ (\pi /2)*sign(\varepsilon (t)),\mathrm{else} \end{array}\right.$$
where $A_{uv}(x,y)$ is the current position of the AUV. $A_{uv}(x(t),y(t))$ is the real-time obstacle avoidance position of the AUV. δ(t) is the angle between the current AUV position and the obstacle avoidance path end connection and the path. $\psi_{d}$ is the desired heading. $\beta_{i}$ is the angle between the obstacle avoidance track point and the true north direction. ε(t) is the distance between the center of the circle and the AUV. α(t) is the angle between the forward-looking vector and the obstacle avoidance path.

7. Based on the obstacle avoidance track points obtained above, we perform the following obstacle avoidance guidance algorithm to correct the position and heading of the AUV to form a safe obstacle avoidance path.
(45)$$\begin{array}{l} \beta_{i}=a\tan2(P_{j,i,guide}(x,y)-A_{uv}(x(t),y(t)))\\ \varepsilon (t)=||P(x_{o},y_{o})-A_{uv}(x(t),y(t))|{|}_{2}-R_{i,\max}\\ \psi_{d}=\beta_{i}-\alpha (t) \end{array}$$
where α(t) is selected as follows.
(46)$$\alpha (t)=\left\{ \begin{array}{l} a\cos(\left|\varepsilon (t)\right|/\Delta ),\left|\varepsilon (t)\right|\leq \Delta \\ 0,\mathrm{else} \end{array}\right.\;\;\;\;\;\;(46)$$
where $P(x_{o},y_{o})$ is the center of the circle where the maximum turning radius $R_{i,\max}$ of the obstacle avoidance track point is located. $\psi_{d}$ is the desired heading. $\beta_{i}$ is the angle between the obstacle avoidance track point and the true north direction. ε(t) is the distance between the center of the circle and the AUV, and α(t) is the angle between the forward-looking vector and the AUV to the center of the circle.

8. End.

#### 4.5.2. Obstacle Avoidance Algorithm Design for Vortex Obstacles

To solve the trap problem caused by the vortex obstacles for the AUV in the actual underwater environment and improve the collision avoidance efficiency of the complex vortex obstacles, the line-of-sight guidance mechanism for predicting the update continuously of the trajectory segment is used, so that the trap problem caused by the obstacle can be overcome. In Figure 11, the AUV enters the trap of the vortex obstacle. Once the FLS detects the vortex obstacle and the obstacle avoidance distance requirement is satisfied, the vortex obstacle avoidance algorithm is activated. According to the data processing algorithm mentioned above, a black obstacle profile is formed, which can overcome the interference of the complex obstacle profile in obstacle avoidance.

The predictive guidance trajectory segment calculation is performed with the partial vortex obstacle information detected by the sonar and the obstacle avoidance guidance trajectory segment obtained in the sonar’s field of view in Figure 11. First, the obstacle data satisfying the obstacle avoidance distance requirements are selected. Then, the current heading angle and position information of the AUV are combined to calculate the predictive guidance track segment map vector. Lastly, all the obstacle data satisfying the obstacle avoidance distance are processed by formula (47) to obtain the predictive guidance obstacle avoidance trajectory segment.
(47)$$\begin{array}{l} P_{ov}=(p_{o}(x_{o},y_{o})-p_{v}(x_{v},y_{v}))\\ \left\{p_{gv}=(x_{n}-P_{ov}(x),y_{n}-P_{ov}(y))|n=1,\ldots m\right\} \end{array}$$
where $P_{ov}$ is the mapping vector. $p_{gv},m$ is the predictive guidance obstacle avoidance track point and effective range of all effective obstacles detected by sonar.

The orange guidance trajectory segment in Figure 11 can be generated by performing the line-of-sight guidance mechanism for predicting the update continuously of the trajectory segment. Then the historical tracking information is recorded by setting the memory unit $D\in R^{n\times 2}$ including the heading and path position information that the AUV has traveled, which is taken as comparative data for the AUV that enters the vortex obstacle, finds the target, and escapes from the trap. By comparing the current predicted data with the historical track information in D, the correct predicted trajectory segment of escaping the trap can be obtained. Lastly, by choosing the correct predictive guidance track, the AUV can escape from the obstacle trap. After the AUV executes the orange line segment guidance trajectory, the FLS will further detect the remaining information of the vortex obstacle. Then we repeat the above steps to continuously obtain the obstacle avoidance predictive guidance trajectory segment, and realize the purpose of collision avoidance and escape the vortex obstacle trap.

Regarding the termination condition of the vortex obstacle collision avoidance, a virtual target point is designed in this case, and the AUV moves toward the direction combing target’s direction (Tolerance deviation from the target angle: $\Delta \in [-15^{\circ },15^{\circ }]$) and the relative distance reduced between the AUV and the target.

#### 4.5.3. Design of the Obstacle Avoidance Algorithm for Dense Convex Obstacles

If the number of obstacles appearing in the field of view of the FLS are more than or equal to two and they suit the range requirements for obstacle avoidance, the environment is considered to be a dense complex obstacle environment.

It is necessary to design a special obstacle avoidance algorithm for the dense obstacle environment because the single obstacle avoidance algorithm and the vortex obstacle avoidance algorithm cannot solve the problem caused by the dense obstacle environment in the current field of view. The multiple obstacle’s data processing in the field of view of FLS is shown in Figure 12. The optimal predictive obstacle avoidance track point can be obtained by combing obstacle avoidance rules and weight functions with the following formula as follows.
(48)$$P_{j,i,guide}=\frac{P_{j,S_{i},\min}(x,y)+P_{j+1,S_{k},\min}(x,y)}{2},\begin{array}{cc} & \left(j=1,\ldots ,m+1;i,k\in [1,80]\right) \end{array}$$
(49)$$G_{uide}=\{P_{j,i,guide}(x,y)|j=1,\ldots ,m+1;i\in [1,80]\}$$
(50)$$G_{ap}=\{l_{i},i=1,\ldots ,m+1\}$$
(51)$$\lambda_{s,i}=\left\{ \begin{array}{l} 0\begin{array}{cc} , & \begin{array}{cc} & G_{ap,i}\leq 4l_{o} \end{array} \end{array}\\ (\frac{\left|G_{ap,i}-4l_{o}\right|}{6l_{o}}\begin{array}{ccc} {)}^{1/2}, & & 4l_{o}\leq G_{ap,i}\leq 6l_{o} \end{array}\\ 1\begin{array}{cccc} , & & & \begin{array}{cc} & \begin{array}{cc} & G_{ap,i}\geq 6l_{o} \end{array} \end{array} \end{array} \end{array}\right.$$
(52)$$\lambda_{c,i}=\rho_{d,i}\cdot \sin(\varphi_{S_{i}})\cdot \gamma_{c}+\lambda_{s,i}$$
(53)$$P_{guide}=f(G_{ap},\lambda_{c,i})$$
where $G_{uide}$ is the possible existing guidance track points of all obstacle gaps. $G_{ap}$ is the spacing of all obstacles. $\lambda_{s,i},\lambda_{c,i}$ is the spacing width influence factor and track point selection factor, respectively. $\gamma_{c}$ is the attenuation factor. $\rho_{d,i},l_{o}$ are the distance between the track point and the AUV and the length of the AUV, respectively. f is the evaluation function of the optimal guidance trajectory point. m is the number of obstacles detected, and, in Figure 12, m=4.

Considering the influence of the length of the turning radius on the safety of the AUV avoidance obstacle, it is unreasonable to select one location in the current gap as the guidance track point when the largest obstacle gap is found. As a result, it will bring the cost of energy and cause the AUV to deviate from the target point. Therefore, a variety of factors should be taken into account when selecting the attenuation factor and evaluation function. It needs to meet the impact of weights in the selection of trajectory points that $\gamma_{c}=0.5$ is selected, and f selects piecewise linear function. In Figure 12, the blue portion of the obstacle is an area that cannot be detected by the sonar in reality. It is impossible to distinguish between the obstacles No. 1 and No. 2 for the AUV because they belong to one area in the field of view of sonar. However, in a real situation, the obstacle is assembled by the number 1 and 2 obstacles. It is undoubtedly beneficial for obstacle avoidance, in reality, to use the corresponding segmentation algorithm to classify different obstacles. The data processing method in Section 3.4 can be used to realize the segmentation of obstacles No. 1 and No. 2 in Figure 12 and obtain the obstacle spacing $l_{2}$ and the predictive guidance point $P_{2,25,guide}$. Therefore, the optional guidance trajectory point of AUV is added to improve the safety and selectivity of obstacle avoidance.

## 5. Simulation Results and Discussions

In the simulation environment of this paper, three kinds of obstacle environments were established to verify that the obstacle avoidance algorithm mentioned above could solve the obstacle avoidance problem in a complex environment and improve the obstacle avoidance efficiency. Obstacle information in the entire simulation environment, including the type, number, and location of obstacles, is generated randomly. The AUV does not know this information in advance. During the process of AUV sailing to the predetermined target point, the obstacle can be detected when the unknown obstacle enters the field of view of the sonar. Then the AUV avoids obstacles in real time, according to the obstacle avoidance algorithm proposed above. Software simulation experiments were carried out in the MATLAB 2014. A 2000 m × 2000 m two-dimensional area was established, which was divided into 400 m × 400 m task sub-areas. In each task sub-region, several grids of equal size were divided, according to unit length to evaluate the cost of AUV’s trajectory when executing the obstacle avoidance algorithm. In three kinds of the obstacle environment, the speed of the AUV when it does not detect the obstacle is 3.5 m/s. The speed of executing the obstacle avoidance algorithm was 2 m/s, and the unit step time was 0.1 s. The proposed obstacle avoidance algorithm was tested and verified in three kinds of obstacle environments, respectively.

### 5.1. Simulation Verification in a Simple Convex Obstacle Environment

This design requires AUV to have a simple convex obstacle environment obstacle avoidance capability to arrive at virtual target points, to verify the effectiveness and efficiency of the obstacle avoidance algorithm in dealing with simple convex obstacle environments. As shown in Figure 13a, the starting position of the AUV is (180, 180), and the heading angle is set to 0° with the two-dimensional y axis as the true north direction. The virtual target point is a red circle, and the center of the circle is (1800, 1800) and the radius is 10. Meanwhile, seven convex obstacles are generated randomly in the two-dimensional environment to form a simple convex obstacle environment. It is stipulated that, when the distance between the AUV and the virtual target point is 10 m, the obstacle avoidance process is ended, and AUV has reached the target and successfully completed the target searching task.

The obstacle avoidance trajectory of APF in a simple, convex obstacle environment is shown in Figure 13b. Clearly, it can be seen that the obstacle avoidance trajectory is not very smooth. Since the selected obstacle avoidance path is not the best path for the current obstacle environment, the cost value of the obstacle avoidance is much higher than that in Figure 13a. The distance between the AUV and the obstacle is relatively close during the obstacle avoidance process, which is likely to increase the risk of the obstacle avoidance failure.

The trajectory of the PSO algorithm used for obstacle avoidance is shown in Figure 13c. It can be seen from Figure 13c that the AUV chooses the path to avoid obstacles as the obstacle avoidance trajectory, which is close to the virtual target point at the same time. However, this also increases the overall cost of the obstacle avoidance process. Compared with Figure 13b, its track is relatively smooth, but its cost increases. However, there is no probability of an increase in the risk of avoiding obstacles. In a word, the obstacle avoidance algorithm proposed is more suitable to the simple obstacle environment.

### 5.2. Simulation Verification in the Vortex Obstacle Environment

The AUV is designed to find the target point in the vortex obstacle and escape from the obstacle to verify the effect of the obstacle avoidance algorithm proposed for escaping from the vortex obstacle environment. The starting position of the AUV is set to (700, 1200), and the heading angle is set to 0° with the two-dimensional y axis as the true north direction. A red circular target point is set in the vortex obstacle with its center being (910, 1010) and the radius is 10. The stop position is set to (1400, 300) after the AUV escapes from the vortex obstacle. Eventually, the AUV enters the vortex obstacle and escapes from the obstacle safely after detecting the target. Then AUV reaches the specified destination. As shown in Figure 14, a concave obstacle environment in a two-dimensional environment composed of complex vortex obstacles is set. It is stipulated that the target point can be marked when the target point appears within the range of 80 m of the AUV sonar’s field of view. It indicates that the AUV has detected the target in the vortex obstacle and completed the obstacle avoidance task successfully and reaches the predetermined end position when the AUV is within the range of 10 m from the end position.

An obstacle avoidance algorithm is set for the vortex obstacle. Figure 14a shows that the AUV enters the obstacle and detects the target and does not fall into the vortex obstacle. The AUV can escape the obstacle at once and quickly reach the designated termination point. The orange trajectory represents the predicted trajectory formed from the detected obstacle data. Additionally, the blue trajectory represents the true obstacle avoidance trajectory of the AUV. The figure shows that the red target point has been marked in green, which indicates that the target point has been successfully detected.

Figure 14b shows the AUV obstacle avoidance trajectory for the return obstacle realized by the APF method. AUV is stuck in a return obstacle and cannot escape from the obstacle because the repulsive force inside the vortex obstacle is large and the gravitational force at the end position is small. AUV only completes the target detection, but cannot escape the obstacle.

Figure 14c shows the AUV obstacle avoidance trajectory formed by the PSO algorithm. As the figure shows, the AUV can detect the target point and escape the obstacle and reach the specified end position. However, it can fall into the obstacle for a period of time before it escapes from the obstacle, which increases the value of the whole obstacle avoidance process.

### 5.3. Simulation Verification in a Dense Obstacle Environment

This design requires AUV to have a dense convex obstacle environment with an obstacle avoidance capability to arrive at virtual target points and to verify the effectiveness and efficiency of the obstacle avoidance algorithm in dealing with a dense convex obstacle environment. As shown in Figure 15, the starting position of the AUV is set to (180, 180), and the heading angle is set to 0° with the two-dimensional y axis as the true north direction. The virtual target point is the red circle with its center at (1800, 1800) and the radius is 10. Fifteen convex obstacles are generated randomly in the two-dimensional environment to form a dense convex obstacle environment. In this experiment, the situation of the obstacle avoidance process ended when the distance between the AUV and the virtual target point is 10 m, which indicates that the AUV has successfully completed the obstacle avoidance task and reached the target.

Figure 15a shows the obstacle avoidance track formed by the PGOA algorithm in a dense convex obstacle environment. The pink circle in the figure indicates the predictive track points. As the figure shows, the AUV chooses a reasonable obstacle avoidance path to reach the virtual target point without bringing an obstacle avoidance risk. Meanwhile, it can maintain the requirement of safe obstacle avoidance distance. In addition, the cost of the entire obstacle avoidance is controlled within 1500.

Figure 15b shows the obstacle avoidance trajectory formed by the APF method. As the figure shows, the AUV does not select an appropriate path as the track to reach the target. Meanwhile, it tends to increase the risk of obstacle avoidance when AUV crosses the obstacle group with relatively small obstacle spacing, which may result in the failure of obstacle avoidance. The path to avoid obstacles does not meet the safe obstacle avoidance distance at the beginning of obstacle avoidance, which does not meet the obstacle avoidance requirements of this paper. In addition, compared with the cost of Figure 15a, there are also many resources that are wasted.

Figure 15c shows the obstacle avoidance path formed by the PSO algorithm. As the figure shows, the path selected by AUV is very close to the obstacle avoidance trajectory of Figure 15a and the path is relatively smooth. It fully meets the safe obstacle avoidance distance. However, it increases the cost of the overall obstacle avoidance because the trajectory to the target is not the most suitable. Moreover, it is similar to the APF obstacle avoidance cost, which is twice the cost of PGOA.

Figure 16 shows the comparison of the cost of the obstacle avoidance process brought by different algorithms in three kinds of obstacle environments. It can be seen from Figure 16 that the PGOA obstacle avoidance algorithm can successfully complete the obstacle avoidance process in different complex obstacle environments, and the cost of the entire obstacle avoidance process is also minimal. APF can adapt to a simple convex obstacle environment and a dense convex obstacle environment. However, its cost is twice as much as the PGOA algorithm, and it cannot handle the vortex obstacle environment. The PSO algorithm can complete obstacle avoidance for the three kinds of obstacle environments, but the cost is very high. In a word, the obstacle avoidance algorithm proposed in this paper has great advantages in dealing with various obstacle environments.

## 6. Conclusions

The PGOA proposed in this paper deals with various complex obstacle environments, which includes complex convex polygon obstacles and complex concave obstacles. This algorithm can handle simple convex obstacles, dense convex obstacles, and vortex obstacles. The AUV’s obstacle avoidance trajectory is also close to smooth. The AUV through various obstacle environments easily reaches the predetermined virtual target point during the whole obstacle avoidance process. Furthermore, the AUV’s obstacle avoidance process has always met the requirements of safe obstacle avoidance distance, and the cost of the entire obstacle avoidance process is less than other traditional algorithms. During the process of facing the vortex obstacles, the AUV can escape the vortex obstacle environment once inside the vortex obstacles by remembering the direction of the path that it has traveled before, which avoids falling into the obstacles and causing an obstacle avoidance process failure. The next step in this paper is to optimize the proposed algorithm to adapt to the AUV pool experiment and extend the algorithm to the complex marine environment where ocean currents and dynamic obstacles exist. At the same time in the actual underwater environment, most obstacles have different shapes at different heights. The obstacle avoidance problem is much more complicated in the three-dimensional environment. Therefore, in the later research, the two-dimensional obstacle avoidance algorithm is used to penetrate into the three-dimensional environment.

## Figures and Tables

**Figure 1 sensors-19-02862-f001:**
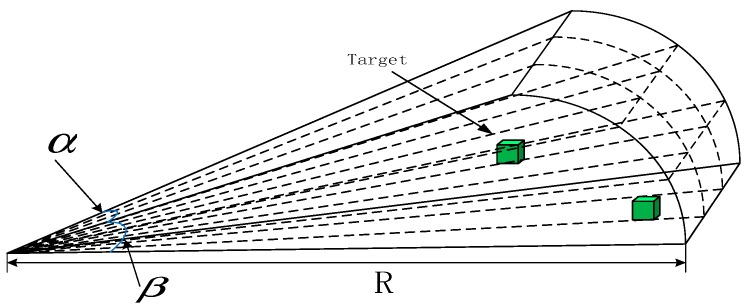
FLS model.

**Figure 2 sensors-19-02862-f002:**
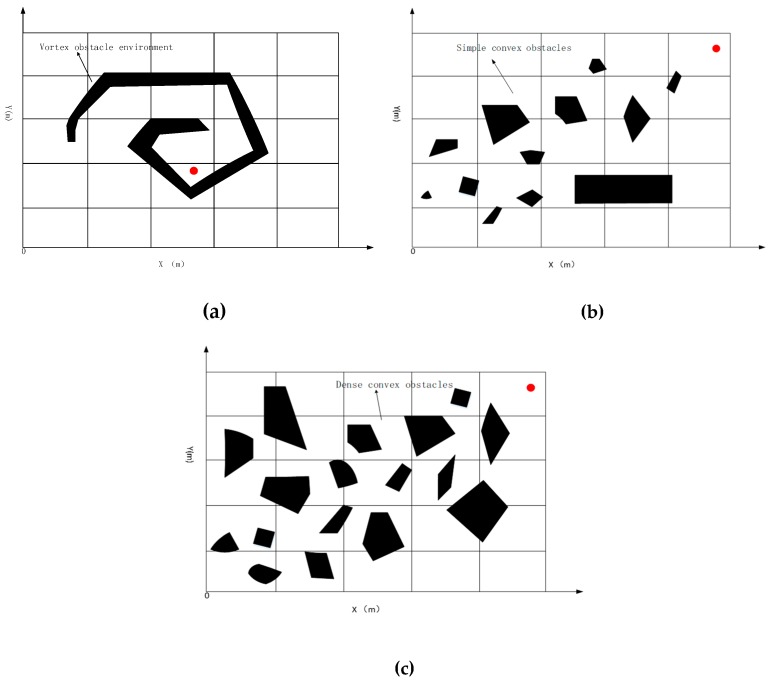
Different types of obstacles. (**a**)No-Convex Obstacles. (**b**) Simple convex obstacle environment. (**c**) Dense Convex Obstacles.

**Figure 3 sensors-19-02862-f003:**
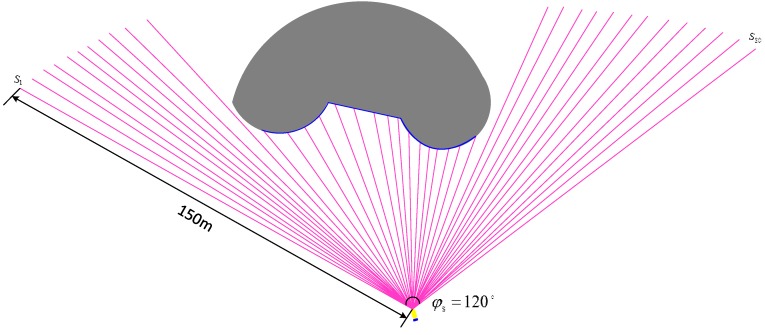
Obstacle detected by a Forward-looking sonar.

**Figure 4 sensors-19-02862-f004:**
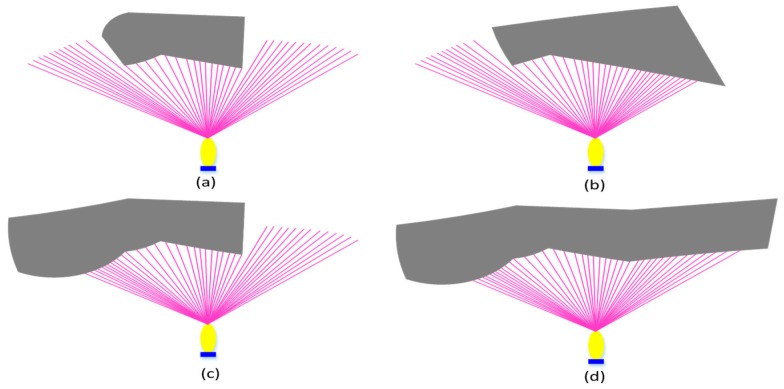
Obstacle classification. (**a**) Bounded obstacle. (**b**) Left bounded obstacle. (**c**) Right bounded obstacle. (**d**) Unbounded obstacle.

**Figure 5 sensors-19-02862-f005:**
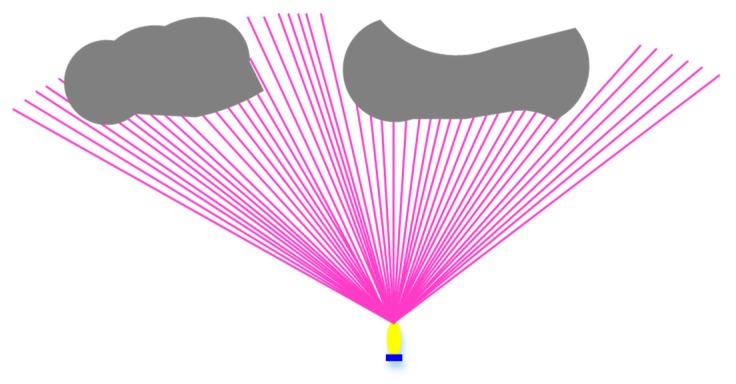
Example of FLS detecting data grouped.

**Figure 6 sensors-19-02862-f006:**
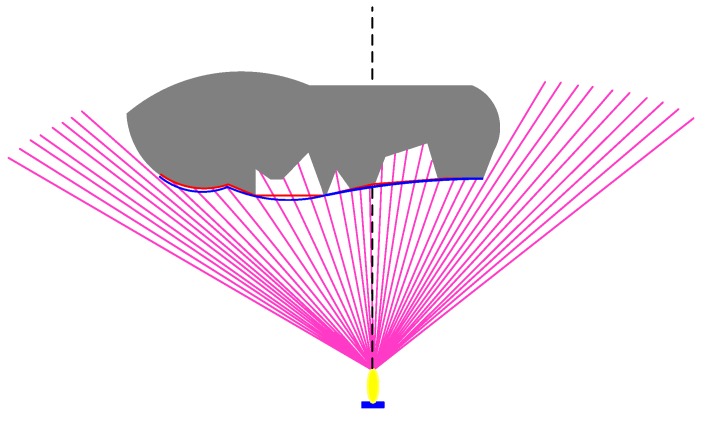
Obstacle surface effect after using the convex algorithm and Bessel interpolation.

**Figure 7 sensors-19-02862-f007:**
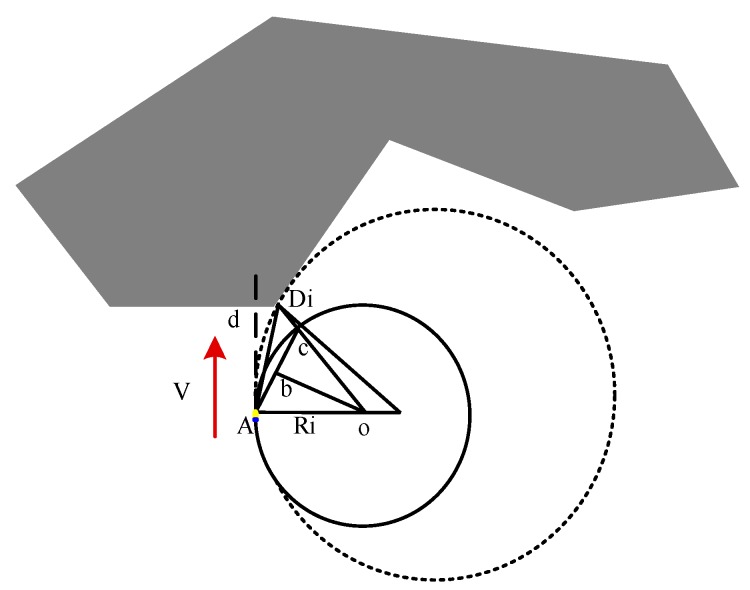
Expected maximum turning radius of the AUV obstacle avoidance point.

**Figure 8 sensors-19-02862-f008:**
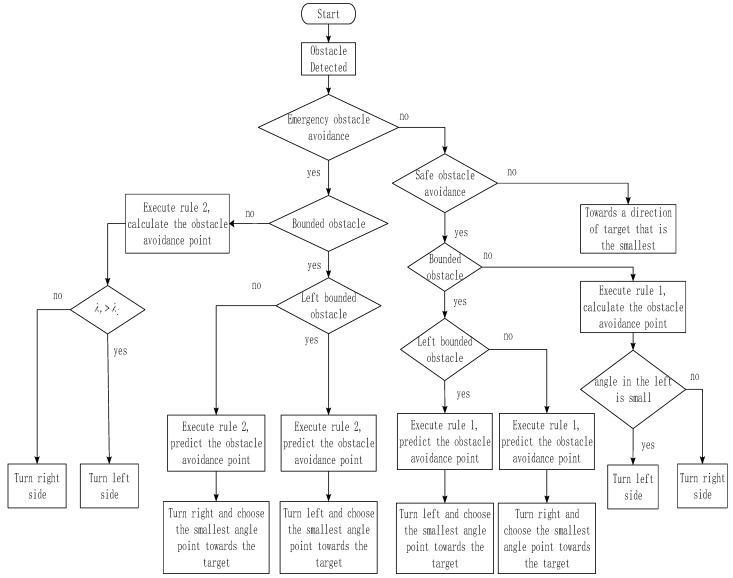
The obstacle avoidance rule shown by a flowchart.

**Figure 9 sensors-19-02862-f009:**
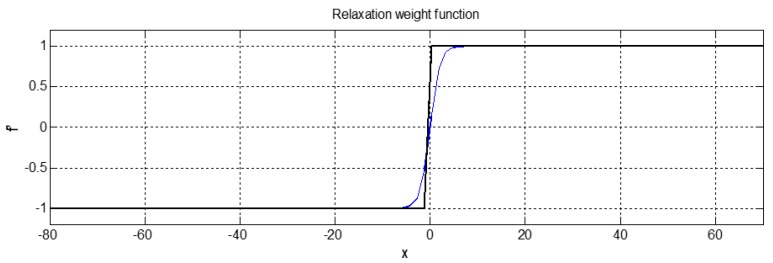
Comparison results of the weight functions before and after relaxation.

**Figure 10 sensors-19-02862-f010:**
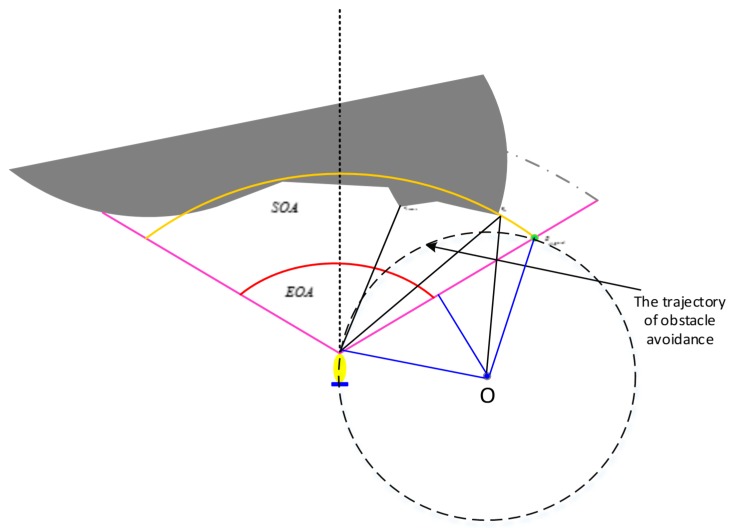
Path design for the right bounded obstacle.

**Figure 11 sensors-19-02862-f011:**
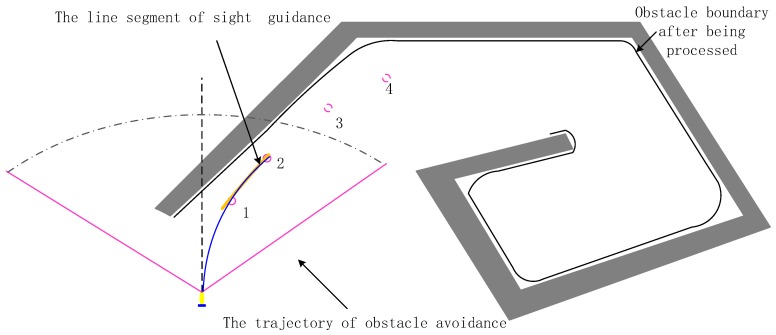
Path design for the obstacle of the vortex obstacle.

**Figure 12 sensors-19-02862-f012:**
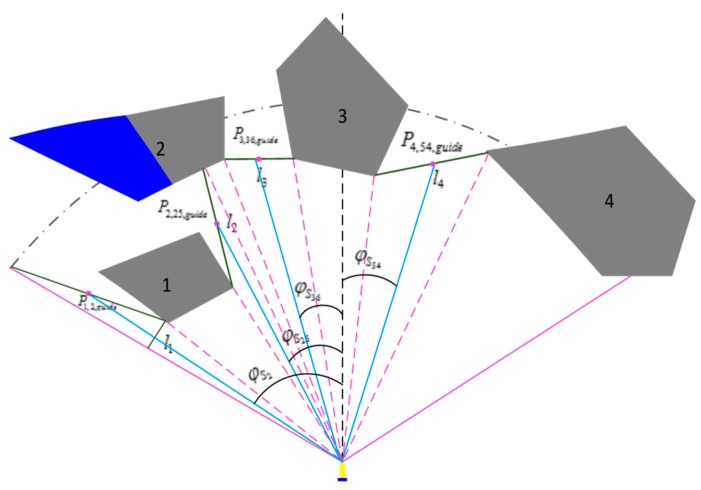
Dense obstacles environment avoidance.

**Figure 13 sensors-19-02862-f013:**
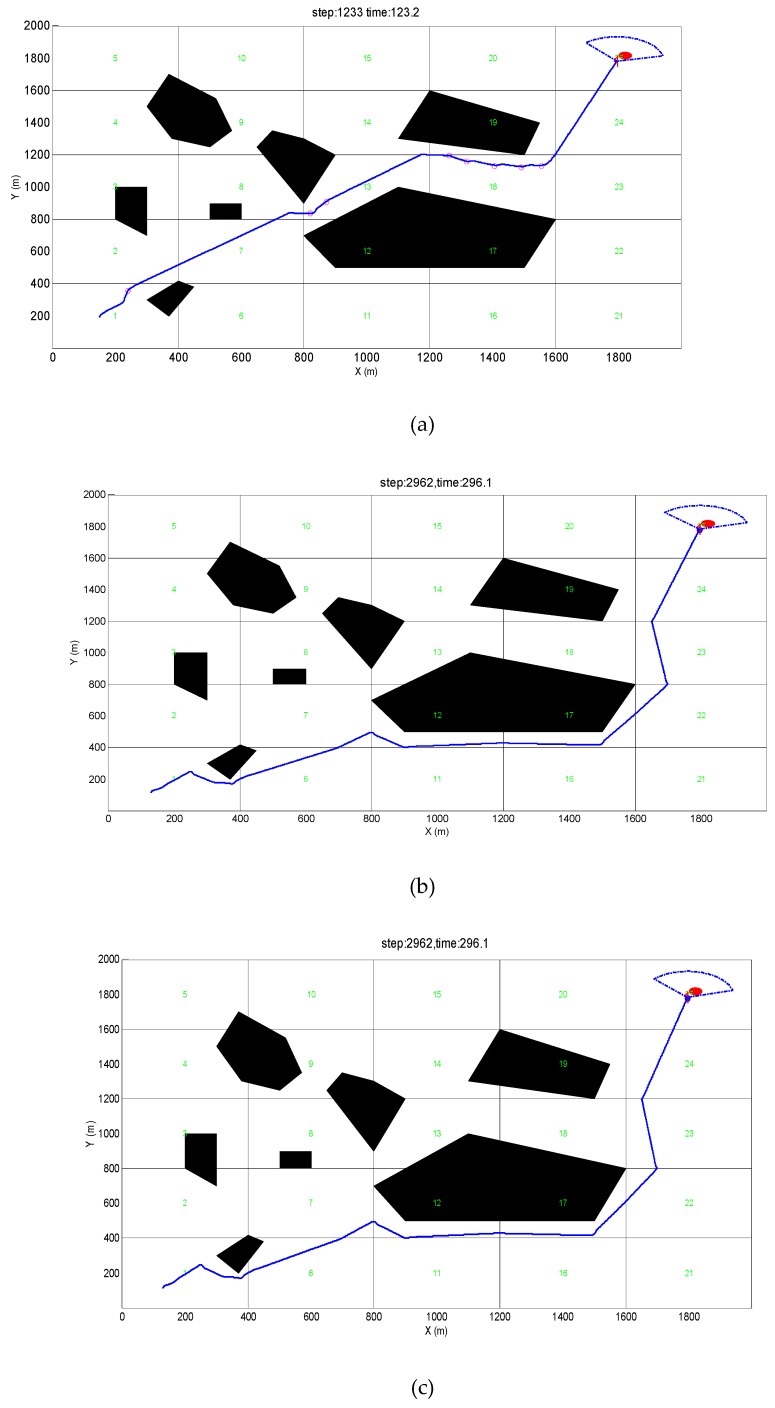
Simple obstacle environment avoidance trajectory. (**a**) PGOA (Predictive guidance obstacle avoidance). (**b**) APF (Artificial potential field). PSO (Particle swarm optimization).

**Figure 14 sensors-19-02862-f014:**
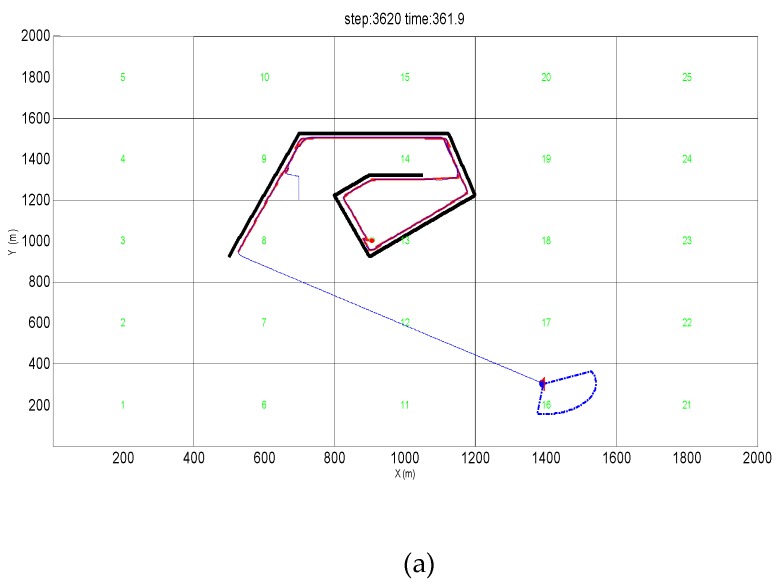
Trap obstacles environment obstacle avoidance trajectory. (**a**) PGOA (Predictive guidance obstacle avoidance). (**b**) APF (Artificial potential field). (**c**) PSO (Particle swarm optimization).

**Figure 15 sensors-19-02862-f015:**
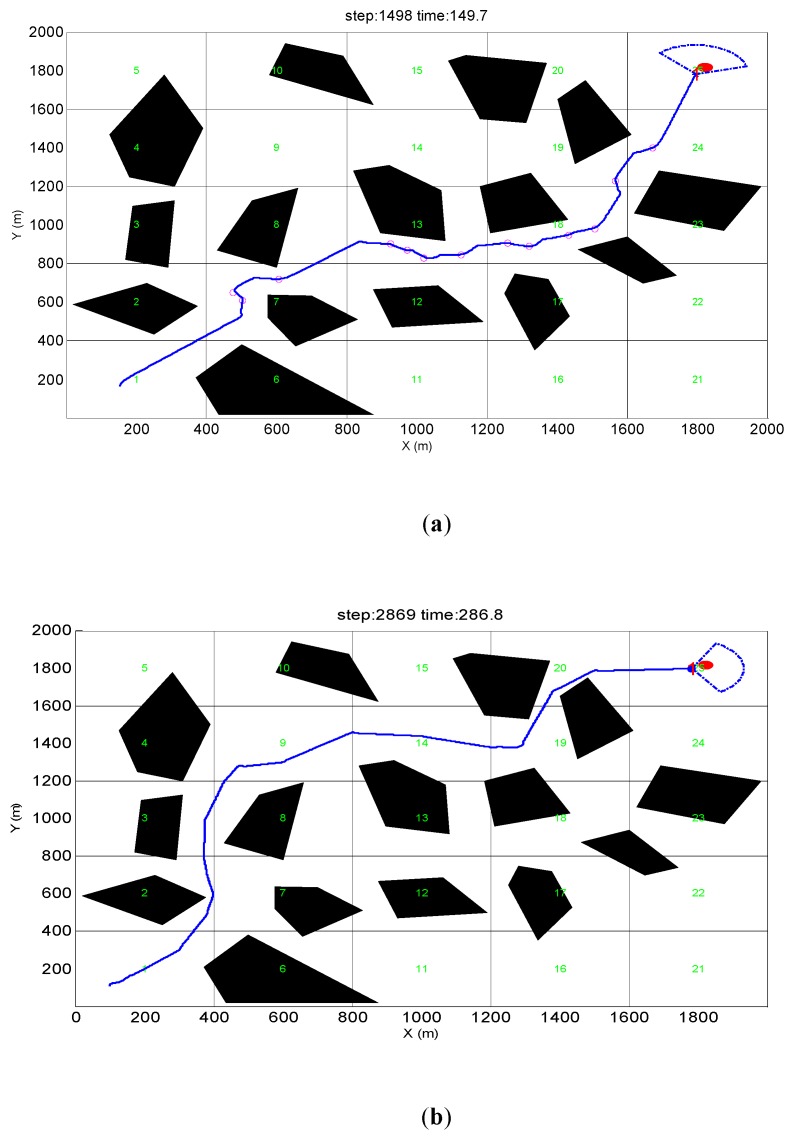
Vortex obstacles environment obstacle avoidance trajectory. (**a**) PGOA (Predictive guidance obstacle avoidance). (**b**) APF (Artificial potential field). (**c**) PSO (Particle swarm optimization).

**Figure 16 sensors-19-02862-f016:**
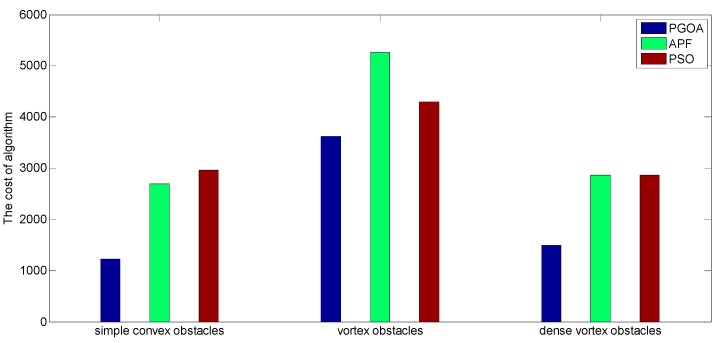
Algorithmic cost in three obstacle environments.

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
