# Peer review of "A Predictive Guidance Obstacle Avoidance Algorithm for AUV in Unknown Environments"

_sensors, 2019, doi:10.3390/s19132862_

Round 1

Reviewer 1 Report

In this paper, the authors proposed methods for AUV obstacle avoidance in unknown environments. It is writen in a good format with clear structure described their ideas and simulated results well. However, there are a few important issues to be cleared as stated below.

1. 'unknown environments' as authors claimed, however, it seems that they assumed the obstacles have already been detected clearly, in their simulation. If so, why state unknown environments? it is acturally talking about obstacle avoidance in a simulated known environment.

2. Lack of clear understanding of the the real challenges in AUV with the current introduction, should add more about real time abstacle detection and avoidance literature review as well to make the whole picture clearer.

3. This paper is about optimization - if taking the AUV etc out, it fits to robots/vehicles/UAVs/any other automonous mobile machines easily. I think there are similar issues however the unique challenges in the AUV application have not been idenfitifed or illustrated well in this paper.

4. The significance of the proposed methods is unclear.

There are many sentences difficult to read, or need rewording or with typo. For example, page1, introduction, sentence starts from line 4 ends in line 5.

Author Response

Dear Reviewer:

I have revised and supplemented the manuscript based on your suggestions, and revised the language expression of the article. The following a separate response to each of your suggestions.

(1)  For the reviewer's opinion (1), the question "AUV has clearly searched obstacles in the simulation experiment of unknown environment" is explained as follows: 1. the global environment is constructed during the simulation, and the obstacle information in the environment including the type, quantity and position of obstacles are randomly generated. AUV does not know this information in advance. Only during the search process, the AUV can judge the obstacle information according to the algorithm proposed in the paper when obstacles entered the field of view of the sonar collection. 2. In the sonar model described in Section 2.3 of the article, the sonar can only process obstacle information that satisfy the obstacle in the effective field of view of the sonar, and obstacles that not appear in the field of view of sonar are unknown obstacles for AUV.

(2)  In the review opinion (2), “lack of a clear understanding of the real challenges of AUV, should add more real-time abstract detection”, explained as follows: when the AUV searches underwater, it basically sneaks into a fixed depth. The sonar sensor is used to detect obstacles in the effective field of view in real time in a state of fixed depth navigation. The process can be abstracted into horizontal motion, and the proposed two-dimensional obstacle avoidance algorithm is effective and feasible. The problem of avoiding the literature review expression mentioned in opinion (2) has been adopted and the first section of the article has been revised and enriched.

(3)  For the question in the review opinion (3), about "the algorithm proposed in the article may not be applicable to robots/vehicles/unmanned aerial vehicles/other mobile robots" is explained as follows: 1. the unique challenge of obstacle avoidance algorithm is described in the first section and 2.1 section; 2. In Section 2.2, the obstacle avoidance algorithm in this paper is aimed at AUV, so the AUV motion model and AUV detection sonar sensor are established, and the characteristics of AUV are simplified in the underwater obstacle avoidance process. Therefore the algorithm can be applied to robots/vehicles/unmanned aerial vehicles/other mobile robots.

(4)  For the question in the review opinion (4) “emphasizing the importance of research methods” has been adopted. The explanation is as follows: The characteristics and importance of the obstacle avoidance algorithm are mainly described in the introduction of Section 1 of the article. The importance of the AUV obstacle avoidance algorithm is also described in Sections 4 and 6 of the article.

Thank you a lot for your suggestions on the article.

Reviewer 2 Report

This paper presents a predictive guidance obstacle avoidance algorithm in unknown environments for autonomous underwater vehicle that adapts to in multiple obstacle environments complex. The contribution and innovation of this paper sounds good. I would like to suggest the minor revision before the acceptance of this paper.

1. The contents of this paper should be refined a bit; the current version is too long.

2. The major contribution of this paper might be added at the end of section ‘1. Introduction’. Too long paragraph could be decomposed into two or three parts.

3. Equation (1) is not clear, please revise it.

4. In the section ‘5.3. Simulation verification in dense obstacle environment’, more contrastive analyses of other related methods should be added in this part.

5. The contents are well organized, one problem in the section ‘1. Introduction’ should be concerned. Several Intelligent obstacle avoidance algorithms could be added in this part. Some literatures are suggested as below:

(1) Linear Quadratic Optimal Control-Based Missile Guidance Law With ObstacleAvoidance. IEEE Transactions on Aerospace and Electronic Systems, 55(1): 205-214, 2019.

(2) Localization, obstacle avoidance planning and control of a cooperative cable parallel robot for multiple mobile cranes. Robotics and Computer-Integrated Manufacturing, 34: 105–123, 2015.

(3) Integration of Hardware and Software Designs for Object Grasping and Transportation by a Mobile Robot With Navigation Guidance Via a Unique Bearing-Alignment Mechanism. IEEE/ASME Transactions on Mechatronics, 21(1): 576-583, 2016.

Author Response

Dear Reviewer:

I have revised and supplemented the manuscript based on your suggestions, and revised the         language expression of the article. The following a separate response to each of your          suggestions.

(1)       For the review comments (1), the question "The content and length of the paper is too long" has been adopted. The explanation is as follows: the content of the article has been deleted, the length has been shortened, and the integrity of the article has been guaranteed.

(2)       For the review comments (2), the question "the major contribution of this paper might be added at the end of section ‘1. Introduction’. Too long paragraph could be decomposed into two or three parts." has been adopted. The explanation is as follows: The first section of the article is adjusted, and the contribution of the article is clearly explained at the end of the first section.

(3)       For the question of the review opinion (3) "the equation (1) is unclear", the explanation is as follows: Since the typesetting problem changes its abbreviation into two formulas, the abbreviated formula matrix of the equation (1) is described in the formula (2).

(4)       For the review opinion (4), “In the section 5.3 more contrastive analyses of other related methods should be added” has been adopted. The description is as follows: In the section 5.3 of the paper, the simulation diagrams of the obstacle avoidance algorithm and other methods are introduced respectively and contrastive analyses.

(5)       For the review comments (5) "Several Intelligent obstacle avoidance algorithms could be added in section 1" has been adopted. The description is as follows: The section 1 of the article has been revised, and the references to relevant references have been added and enrich the research content of the related fields of in the introduction.

Thank you a lot for your suggestions on the article.

Reviewer 3 Report

The paper presents a predictive guidance obstacle avoidance algorithm for AUV in unknown environments.

The topic of the paper is interesting and relevant to the aims of the journal.

The algorithm is deeply described, and the results well presented.

Therefore, I have some minor comments that need to be addressed to improve the manuscript.

1.       English should be improved, and the manuscript should be checked for typos.

2.       The authors should better highlight the novel contributions of this work in the introduction.

3.       The authors should check the paper template: in some pages the line and paragraph spacing changes.

Author Response

Dear Reviewer:

I have revised and supplemented the manuscript based on your suggestions, and revised the language expression of the article. The following a separate response to each of your suggestions.

(1)     For the review comments (1) the suggestion of "English should be improved, and the manuscript should be checked for typos" has been adopted, and the English expression of the article as a whole has been revised based on the comments of the reviewer.

(2)     For the review comments (2) “the authors should better highlight the novel contributions of this work in the introduction” has been adopted. The explanations are as follows: in the first section of the article, the research focus and outstanding contributions of the article are presented. In section 2.1 of the article, the difficult problems solved in the study are mentioned. Finally, in the conclusion of the sixth section of the article, the role and significance of this work in related fields are proposed.

(3)     For the review comments (3) about "the article template and paragraph", has been adopted, the paragraph layout of the article has been sorted out and revised.

Thank you a lot for your suggestions on the article.

Reviewer 4 Report

The manuscript „A Predictive Guidance Obstacle Avoidance Algorithm for AUV in Unknown Environments” presents interesting topic because significance of OA systems for AUV will increase in near future.

One of the main disadvantages of the algorithm that it was built on the 2-D environment. In practice most of the obstacles have different shape at different heights. Solution of obstacle avoidance would be much more complicated in 3-D environment. I am aware that it is not the aim of the study, but in my opinion short proposal of development of this algorithm for 3-D environment should be presented.

Another disadvantage of the obstacle avoidance algorithm is lack of considering various size of the AUV. In case of bigger size of AUVs it cannot work properly because not only size but limitation of navigation and delay in changing the trajectory of bigger AUVs, too.

There are many studies on similar topic (e.g. https://www.mdpi.com/1424-8220/18/2/438, https://www.mdpi.com/1424-8220/19/1/20, https://doi.org/10.1007/s10514-015-9532-2). Novelty of the proposed algorithm in comparison to already existed algorithms should be highlighted.

Please double-check all the manuscript to remove typos, e.g. line 214: “In For…”

Author Response

Dear Reviewer:

I have revised and supplemented the manuscript based on your suggestions, and revised the language expression of the article. The following a separate response to each of your suggestions.

(1)     For the review comments (1) about "The author should increase the simple proposal to study the method into a three-dimensional environment" has been adopted. When the AUV searches underwater, it basically sneaks into a fixed depth. The sonar sensor is used to detect obstacles in the effective field of view in real time in a state of fixed depth navigation. The process can be abstracted into horizontal motion, and the proposed two-dimensional obstacle avoidance algorithm is effective and feasible. Section 6 of the article is modified, and AUV obstacle avoidance in the three-dimensional environment will be proposed in the future study.

(2)     For the review comments (2) “the authors should consider that in case of bigger size of AUVs it cannot work properly because not only size but limitation of navigation and delay in changing the trajectory of bigger AUVs” the explanations are as follows: The focus of the research is on how to implement an effective obstacle avoidance algorithm in the unknown environment based on the AUV motion model, which has certain reference value for different types of AUV obstacle avoidance. The characteristics of motion flexibility and navigation are affected by the AUV drive mechanism and configuration during the actual experiment process. A small modification can be made for different objects, which is completely applicable to the large AUV obstacle avoidance process.

(3)     For the review opinion (3),”Novelty of the proposed algorithm in comparison to already existed algorithms should be highlighted” has been adopted. We have modified the section 1 of the article and compared the relevant research fields. The novelty of the proposed algorithm has been highlighted.

(4)     Regarding the review comments (4), the suggestion of "checking all the manuscripts to remove misspellings" has been adopted, and the entire article has been revised for the errors in the writing.

Thank you a lot for your suggestions on the article.

Round 2

Reviewer 1 Report

The authors have not replied my questions raised earlier. In short, in an exremly simplified simulated environments for AUVs, what are the 'unknow'? if 'unknow', how those unknown be used for global optimal results? Please list your answer to the questions raised.

Author Response

Dear reviewer:

       I have uploaded a response to your review comments.

Reviewer 4 Report

In my opinion, the manuscript has been corrected by authors enough for its acceptance.

Author Response

(The authors gave the same response as above.)

Round 3

Reviewer 1 Report

Further proof reading required.